# Dynamic Gaussian Splatting from Defocused and Motion-blurred Monocular Videos

**Xuankai Zhang[1], Junjin Xiao[1], Qing Zhang[1]***
[1]Sun Yat-sen University
zhangxk53@mail2.sysu.edu.cn,xiaojj37@mail2.sysu.edu.cn,
zhangqing.whu.cs@gmail.com
https://dydeblur.github.io/

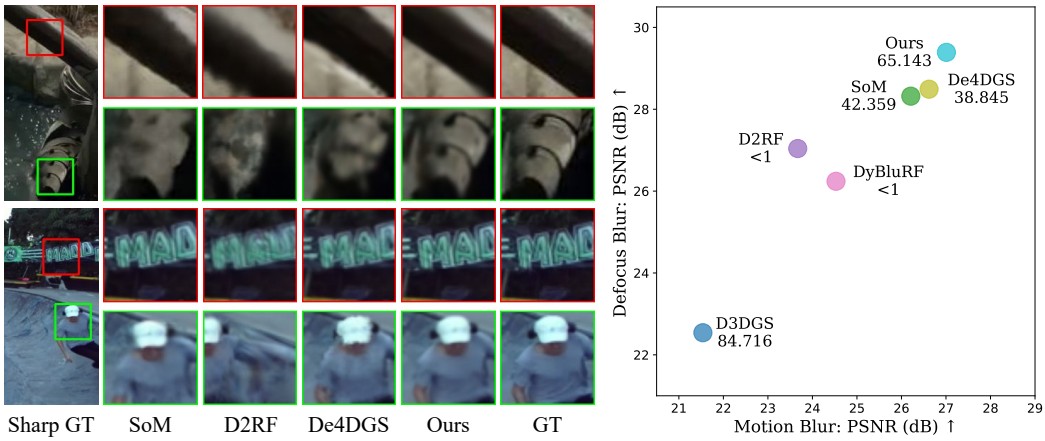

Figure 1: **Performance of our method.** Our method allows to synthesize high-quality sharp novel views for videos with defocus blur (top) and motion blur (bottom). As shown on the right, our method not only obtains significantly better results than existing methods, e.g., D3DGS [15], SoM [52], D2RF [32], DyBluRF [47], and De4DGS [58], but also achieves a performance of 65.143 FPS at a resolution of $512 \times 288$ on an NVIDIA RTX 3090 GPU.

## Abstract

This paper presents a unified framework that allows high-quality dynamic Gaussian Splatting from both defocused and motion-blurred monocular videos. Due to the significant difference between the formation processes of defocus blur and motion blur, existing methods are tailored for either one of them, lacking the ability to simultaneously deal with both of them. Although the two can be jointly modeled as blur kernel-based convolution, the inherent difficulty in estimating accurate blur kernels greatly limits the progress in this direction. In this work, we go a step further towards this direction. Particularly, we propose to estimate per-pixel reliable blur kernels using a blur prediction network that exploits blur-related scene and camera information and is subject to a blur-aware sparsity constraint. Besides, we introduce a dynamic Gaussian densification strategy to mitigate the lack of Gaussians for incomplete regions, and boost the performance of novel view synthesis by incorporating unseen view information to constrain scene optimization. Extensive experiments show that our method outperforms the state-of-the-art methods in generating photorealistic novel view synthesis from defocused and motion-blurred monocular videos. Our code is available at https://github.com/hhhdddddddd/dydeblur.

---

*Corresponding author.

39th Conference on Neural Information Processing Systems (NeurIPS 2025).

# 1 Introduction

Novel view synthesis of dynamic scenes from monocular videos is a very important problem, with applications in various scenarios such as augmented reality, virtual reality, and 3D content creation. Recent progress in this field mostly aims to learn renderable 3D Gaussian representations from monocular videos. Although some of them have demonstrated impressive results of novel view synthesis [24, 45, 52, 63, 53], their performance typically deteriorates significantly on blurry videos, making methods applicable to blurry monocular videos particularly necessary.

Some recent variants [58, 47, 30, 4, 32] of 3D Gaussian Splatting (3DGS) [15] and Neural Radiance Field (NeRF) [35] have attempted to reconstruct dynamic scenes from blurry monocular videos, where [58, 47, 30, 4] tackles motioned-blurred videos by computing a weighted sum of multiple rendered images within the exposure period, while [32] focuses on dealing with defocus blur using layered Depth-of-Field (DoF) volume rendering. Although these methods demonstrate promising results, due to the significant difference between the formation process of motion blur and defocus blur, their effectiveness is limited to either motion-blurred or defocused videos. Currently, there does not exist a method that can effectively handle both types of blurry videos while enabling high-quality novel view synthesis.

In this work, we present a framework that allows high-quality dynamic Gaussian Splatting from both defocused and motion-blurred monocular videos. To this end, we employ blur kernel based convolution to jointly model the two blur types. To obtain reliable blur kernels from dynamic scenes, we develop a blur prediction network (BP-Net) that is subject to a blur-aware sparsity constraint to simultaneously predict the blur kernel and pixel-level intensity. Moreover, we introduce a dynamic Gaussian densification strategy to mitigate the lack of Gaussians for incomplete regions, and propose to boost the performance of novel view synthesis by incorporating unseen view information to constrain scene optimization. In summary, our main contributions are as follows:

- We introduce a unified framework for dynamic Gaussian Splatting from both defocused and motion-blurred monocular videos, which to our knowledge, makes the first attempt in this field.
- We develop a blur prediction network equipped with a blur-aware sparsity constraint, and introduce a dynamic Gaussian densification strategy as well as a unseen view combined scene optimization scheme.
- We show that our method outperforms previous methods on both defocused and motion-blurred monocular videos.

# 2 Related Works

**4D Reconstruction.** Recent works have extended 3DGS to 4D domain to model dynamic scenes [11, 64, 8, 29, 63, 26]. One line of works define motion as a time-conditioned deformation network that warps Gaussians from canonical space to observation space [57, 63, 7, 12, 16, 13, 27]. Among these, DeformableGS [63] employs MLPs to predict Gaussian deformations, while 4DGaussians [57] replaces MLPs with a multi-resolution HexPlane [5]. Another line of works model motion as trajectories of 3D Gaussians [31, 26, 40, 14, 2, 52, 48, 28]. E-D3DGS [2] introduces per-Gaussian and temporal embeddings to encode time-aware information, whereas Shape-of-Motion [52] models motion as a linear combination of $\mathbb{SE}(3)$ motion bases. However, existing dynamic 3DGS methods rely on sharp input images and often struggle to synthesize photorealistic novel views when motion or defocus blur is present in the inputs.

**Image deblurring.** Early works on image and video deblurring [19, 20, 49, 36, 65] primarily rely on CNNs [46] and RNNs [69]. Later approaches enhance the deblurring process by incorporating additional cues such as depth maps [39, 25, 50, 1], light fields [43, 44], and 3D geometry [37, 38, 55, 59, 61]. Leveraging multi-view and 3D information, NeRF- and 3DGS-based deblurring methods can generate sharp images with improved scene consistency [41, 21, 34, 22, 23, 9]. Among these, some approaches [34, 22, 33, 3, 6] model blur by predicting deformable sparse kernels with per-pixel bases. Others simulate the physical formation of blur by integrating multiple sharp images over the exposure time [68, 51, 38] or use a camera model that learns aperture and focal length parameters [55, 59]. Additionally, some works [47, 32, 58, 30, 4] incorporate 3D geometry into monocular video deblurring. However, existing methods are typically designed for either defocus blur or motion blur.

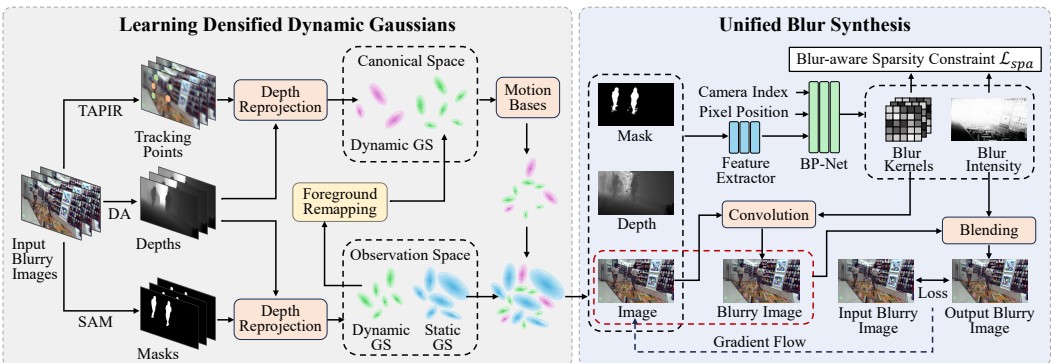

Figure 2: **Overview of our method.** We initialize static Gaussians via depth reprojection and dynamic Gaussians from tracking points, modeling their transformations with learnable motion bases. After stable training, we densify dynamic Gaussians using foreground remapping. For blur modeling, a network predicts per-pixel blur kernels and intensity, enabling blur synthesis through convolution and blending operation. The reconstruction loss between synthesized and input blurry images optimizes the Gaussians for sharper results.

In contrast, our approach effectively removes both motion blur and defocus blur from monocular videos, producing high-quality novel-view images of dynamic scenes.

## 3 Method

Our method aims to achieve dynamic Gaussian Splatting that enables high-fidelity yet sharp novel view synthesis from any given monocular video with defocus blur or motion blur. Figure 2 presents the overview of our method.

### 3.1 Learning Densified Dynamic Gaussians

**Representing scene as dynamic 3D Gaussians.** We adopt Shape-of-Motion [52] to initialize 3D Gaussians and model scene motion. It separately models dynamic and static Gaussians, representing motion with a compact set of $\mathbb{SE}(3)$ motion bases. Specifically, dynamic Gaussians are initialized by reprojecting the depth of 2D tracking points in the canonical frame $t_0$, obtained via Depth-Anything [62] and TAPIR [10]. Static Gaussians are initialized by reprojecting background depth across frames and share the same parameter composition as the original 3D Gaussians. Dynamic Gaussians include an additional motion coefficient $\mathbf{w} \in \mathbb{R}^{N_b}$, which combines $\mathbb{SE}(3)$ motion bases to model their motion. The affine transformation from the canonical frame $t_0$ to the observation frame $t$ is computed as follows:

$$\mathrm{T}_{t_0 \to t} = \sum_{b=0}^{N_b} \mathbf{w}^{(b)} \mathrm{T}_{t_0 \to t}^{(b)}, \tag{1}$$

Where $N_b$ denotes the number of $\mathbb{SE}(3)$ motion bases, and $\mathrm{T}_{t_0 \to t}^{(b)}$ represents the affine transformation of motion base $b$. The affine transformation $\mathrm{T}_{t_0 \to t}$ consists of two components: rotation matrix $\mathrm{R}_{t_0 \to t}$ and translation vector $\mathrm{t}_{t_0 \to t}$. The transformation of the pose parameters $(\mu, R)$ for the dynamic Gaussian from the canonical frame $t_0$ to the observation frame $t$ is defined as follows:

$$\mu_t = \mathrm{R}_{t_0 \to t} \mu_{t_0} + \mathrm{t}_{t_0 \to t}, \quad \mathrm{R}_t = \mathrm{R}_{t_0 \to t} \mathrm{R}_{t_0}. \tag{2}$$

The transformed dynamic Gaussians in observation space, along with the static Gaussians, are then processed through a differentiable rasterization pipeline to generate the final rendered image $\tilde{I}$, depth map $\tilde{D}$, and mask $\tilde{M}$.

**Dynamic Gaussian densification.**

Shape-of-Motion [52] initializes dynamic Gaussians using point clouds by reprojecting 2D tracking points from the canonical frame into 3D space. Although the canonical frame is selected as the one containing the most visible 2D tracking points across all frames, tracking points that are invisible in

the canonical frame can still lead to partial dynamic Gaussians with inaccurate depth. This occurs because the depth values at the corresponding locations of invisible 2D tracking points do not reflect the true depth after reprojection. To address this, we initialize dynamic Gaussians using only visible 2D tracking points in the canonical frame. While this improves initialization accuracy, it may lead to incomplete dynamic regions. To compensate, we supplement dynamic Gaussians by reprojecting dynamic regions with depth maps from all observation frames. Then, we transform the dynamic Gaussians in the observation frames to the canonical frame using foreground remapping. Since this operation requires relatively stable and accurate motion bases, we perform dynamic Gaussian densification only once after the first $N_d$ training iterations.

To this end, we first identify dynamic pixels in the training images using motion masks $M$, obtained via the off-the-shelf method SAM [18]. A random subset of pixels is selected in each observation frame. For a dynamic pixel $g$ in frame $t$, the mean $\mu_t^g$ of the corresponding Gaussian $G$ in observation space is defined as:

$$\mu_t^g = P_t^{-1}(\pi_t^{-1}(g, D(g))), \tag{3}$$

where $D(g)$ is the depth of pixel $g$, $\pi_t$ is the projection function for frame $t$, and $P_t$ represents the camera extrinsics for frame $t$. We use foreground remapping to compute the corresponding position $\mu_{t_0}^g$ of Gaussian $G$ in the canonical frame. Specifically, we first determine all affine transformations for all existing dynamic Gaussians from $t_0$ to $t$. We then select the affine transformation $T_{t_0 \to t}^{G'} = [R_{t_0 \to t}^{G'}, t_{t_0 \to t}^{G'}]$ corresponding to the dynamic Gaussian $G'$ that is closest to $\mu_t^g$ after transformation, which gives:

$$\mu_{t_0}^g = (R_{t_0 \to t}^{G'})^{-1}(\mu_t^g - t_{t_0 \to t}^{G'}). \tag{4}$$

## 3.2 Unified Blur Synthesis

During training, we explicitly model the blurring process and jointly optimize a sharp 3DGS representation along with the blur parameters, ensuring that the synthesized blurry images match the input blur image. Although the formation processes of defocus blur and motion blur are fundamentally different, both types of blur can be approximated as a weighted combination of a pixel and its neighboring pixels. Thus, the generation processes of motion blur and defocus blur can be unified under a simple yet powerful mathematical model:

$$\tilde{B}(x) = \sum_{x_i \in \mathcal{N}(x)} \tilde{I}(x_i) k_x(x_i) \text{ s.t} \sum_{x_i \in \mathcal{N}(x)} k_x(x_i) = 1, \tag{5}$$

where $\tilde{I}(x_i)$ is the clear pixel value at pixel coordinates $x_i$ around the neighborhood of $x$, and $k_x$ denotes the blur kernel for pixel $x$.

Using the per-pixel blur kernel $k_x$ from Eq. (5), we can generate a blurry image $\tilde{B}$ from the sharp image $\tilde{I}$. The main challenge is estimating a reasonable blur kernel $k_x$ for each pixel $x$. An intuitive solution would be to use a CNN to estimate $k_x$ for each pixel in the sharp image $\tilde{I}$ obtained from rasterization. However, jointly optimizing the 3DGS and the CNN blur kernel prediction network under the constraint of blurry images $B$ can lead to non-rigid distortions in the reconstructed 3DGS. This is in line with expectations because it is possible that the reconstructed 3DGS and the CNN blur kernel prediction network deform together without affecting the reconstructed blurry result. To address this, we design the Blur Prediction Network (BP-Net) $F_\Theta$, which simultaneously predicts the per-pixel blur kernel $k_x$ and the corresponding blur intensity $m_x$. Using the blur intensity $m_x$, we compute the output blurry image pixel value $\hat{B}(x)$ by blending the sharp image pixel value $\tilde{I}(x)$ and the blurry image pixel value $\tilde{B}(x)$:

$$\hat{B}(x) = (1 - m_x) \cdot \tilde{I}(x) + m_x \cdot \tilde{B}(x). \tag{6}$$

Computing the reconstruction loss between the output blurry image $\hat{B}$ and the input blurry image $B$ directly constrains the rendered image $\tilde{I}$ using the sharp regions in $B$, progressively guiding it toward a sharper 3DGS-based scene representation, while ensuring that most of the blur is explicitly modeled by the CNN blur kernel prediction network.

**Blur prediction network.** The Blur Prediction Network (BP-Net) is a four-layer CNN model that takes camera and scene information as inputs, both of which significantly contribute to blur

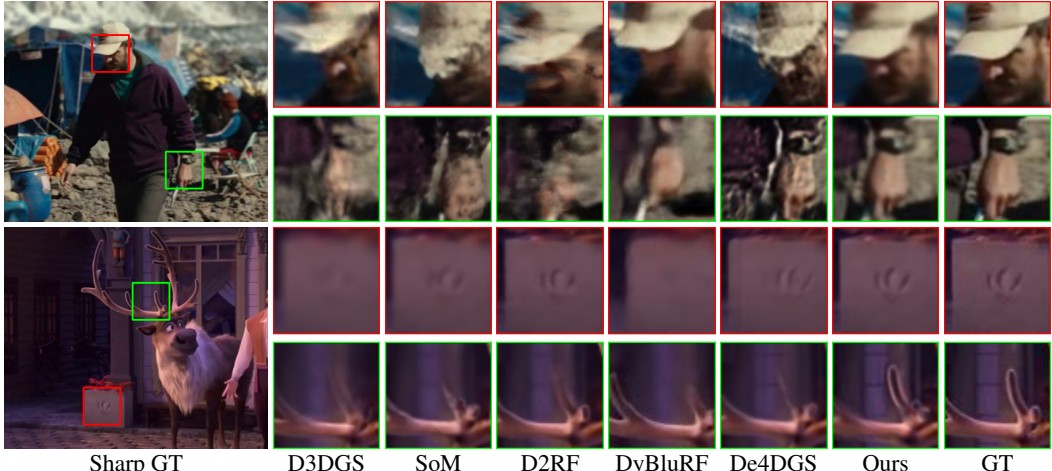

| Sharp GT | D3DGS | SoM | D2RF | DyBluRF | De4DGS | Ours | GT |

Figure 3: **Visual comparison of novel view synthesis on the D2RF defocus blur dataset [32].**

in monocular videos. Camera motion and suboptimal camera settings can introduce blur, while dynamic objects and scene depth strongly correlate with blur magnitude. The camera information is represented by a learnable embedding vector $e(i)$ obtained from the camera view $i$. The scene information is encoded into $f_{scene}$ through a three-layer CNN Scene Feature Extractor, which uses the rendered image $\tilde{I}$, depth $\tilde{D}$, and motion mask $\tilde{M}$ as inputs. Given the variation in blur across different pixels in real blurry images, we also include the pixel coordinate positional encoding $p(x)$ as input to the BP-Net:

$$k_x, m_x = F_\Theta(e(i), f_{scene}(x), p(x)), \tag{7}$$

where $k_x$ and $m_x$ denote the blur kernel and blur intensity for pixel $x$, respectively. To improve the scene information representation, we add skip connections from $f_{scene}$ after the first two layers of the BP-Net.

**Blur-aware sparsity constraint.** The blur kernel weights for mildly blurred pixels should be more concentrated around the center, while those for severely blurred pixels should be more uniformly distributed. To prevent unrealistic blur kernels from excessively blurring mildly blurred regions, we use the blur intensity $m_x$ to constrain the weight distribution of the corresponding blur kernel $k_x$. Specifically, we use the blur kernel center weight $k_x(c)$ to quantify the sparsity of the blur kernel $k_x$. A smaller $k_x(c)$ suggests a more dispersed kernel, implying that the corresponding pixel is more severely blurred. Based on this, we design a blur-aware center weight $c_x$ for the blur kernel $k_x$ as follows:

$$c_x = \text{sigmoid}(scale \cdot (1 - \text{sg}(m_x))), \tag{8}$$

where $scale$ is a scale factor (set to 5), and $\text{sg}$ denotes the stop-gradient operation. Clearly, the pixel-wise blur intensity $m_x$ is negatively correlated with the corresponding blur-aware center weight $c_x$; as $m_x$ increases, $c_x$ decreases. We then compute the $\mathcal{L}_1$ loss between the blur-aware center weight $c_x$ and the blur kernel center weight $k_x(c)$:

$$\mathcal{L}_{spa} = \mathcal{L}_1(c_x, k_x(c)). \tag{9}$$

During training, we adopt the blur-aware sparsity constraint $\mathcal{L}_{spa}$ for training only when the blur intensity $m_x$ has been trained for a number of $N_{spa}$ iterations to be stable.

### 3.3 Training Strategy

Monocular reconstruction of complex dynamic scenes is ill-posed and prone to local minima due to the limited information provided by monocular videos, where each timestamp captures only a single training view. This lack of sufficient views makes it difficult to accurately learn the 3D structure of dynamic scenes, often leading to overfitting to the training views. To mitigate overfitting and extract more cues from monocular videos, we obtain potential appearance information for unseen views surrounding the training views. During training, we use this appearance information to supervise the optimization process every $N_u$ iterations. Since the appearance information from unseen views is

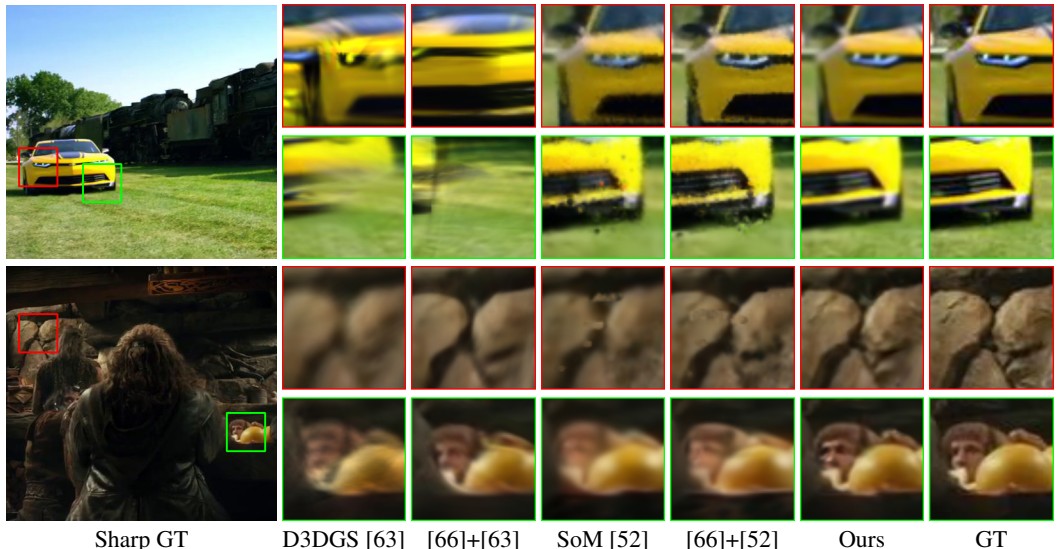

| Sharp GT | D3DGS [63] | [66]+[63] | SoM [52] | [66]+[52] | Ours | GT |

Figure 4: **Visual comparison of novel view synthesis on the D2RF defocus blur dataset [32].** Here, we also compare with methods fed with deblurred images produced by a state-of-the-art video deblurring method [66] to manifest the effectiveness of our method.

less accurate than that from training views, we apply slightly different loss functions for them, as detailed in Section 3.4.

We leverage the geometry and appearance information from the training views to obtain the appearance of unseen views. For instance, consider a pixel $p_s$ in the training view $V_s$. The corresponding pixel $p_t$ in the unseen view $V_t$ can be expressed as:

$$p_t = KP_t^{-1}P_s D_s(p_s)K^{-1}p_s, \tag{10}$$

Where $K$ is the camera intrinsic matrix, $D_s$ is the depth map for the training view $V_s$, $P_s$ and $P_t$ are the camera extrinsics for the training and unseen views, respectively. We then use the color $B_s(p_s)$ of pixel $p_s$ to derive the color $B_t(p_t)$ of the corresponding pixel $p_t$ in the unseen view via reversed bilinear sampling [60, 54]. Similarly, the motion mask $M_t(p_t)$ in the unseen view can be derived from $M_s(p_s)$.

To ensure the accuracy of appearance information in unseen views, we restrict the selection to unseen views close to the training views. In most monocular videos, the camera poses of training views typically form a sequence of consecutive poses directed towards the scene. Therefore, we select unseen views on both sides of the training view sequence and insert unseen views between adjacent training views. We implement this by generating two types of unseen views: (i) parallel-unseen views: generated by interpolating between adjacent training views along the camera trajectory, (ii) perpendicular-unseen views: generated by first computing a local perpendicular direction to the camera trajectory and then perturbing the training view's camera center along this perpendicular direction by a distance of [0.5, 1] (normalized units). Importantly, the timestamp of an unseen view matches the timestamp of the training view from which it is generated.

### 3.4 Loss Function

**Reconstruction loss.** We supervise the training process with a reconstruction loss to align per-frame pixel-wise color input $\mathbf{B}$. We compute the blurry image $\hat{\mathbf{B}}$ according to Eq. (5) and Eq. (6). The blurry image $\hat{\mathbf{B}}$ is supervised by the following reconstruction loss:

$$\mathcal{L}_{rec} = (1 - \beta)\mathcal{L}_1(\hat{\mathbf{B}}, \mathbf{B}) + \beta\mathcal{L}_{ssim}(\hat{\mathbf{B}}, \mathbf{B}), \tag{11}$$

where $\mathcal{L}_1$ and $\mathcal{L}_{ssim}$ are the $_1$ loss and SSIM [56] loss, respectively, and $\beta$ is set to 0.2. We also constrain the scene geometry using the depth $\tilde{\mathbf{D}}$ and mask $\tilde{\mathbf{M}}$:

$$\mathcal{L}_{geo} = \lambda_{depth}\mathcal{L}_1(\tilde{\mathbf{D}}, \mathbf{D}) + \lambda_{mask}\mathcal{L}_1(\tilde{\mathbf{M}}, \mathbf{M}), \tag{12}$$

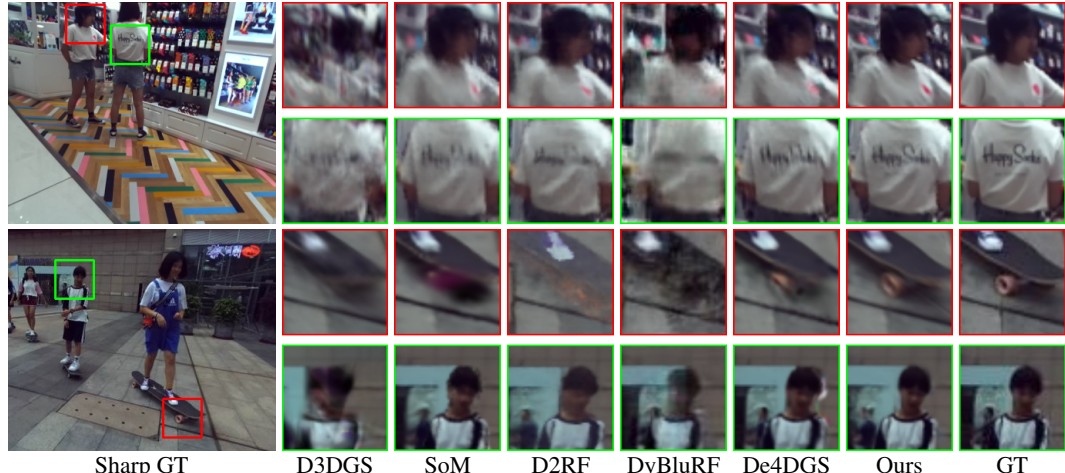

|  | Sharp GT | D3DGS | SoM | D2RF | DyBluRF | De4DGS | Ours | GT |

Figure 5: **Visual comparison of novel view synthesis on the DyBluRF motion blur dataset [47].**

where $\lambda_{depth}$ and $\lambda_{mask}$ are set to 0.075. During iterations with unseen views, we only use the mask, excluding depth, to prevent the inaccurate geometry of unseen views from distorting the scene geometry.

Table 1: **Quantitative comparison of novel view synthesis on the D2RF defocus blur dataset [32] and the DyBluRF motion blur dataset [47].**

| Method | Defocus Blur | | | Motion Blur | | | Param. | Training Time |
|--------|-------|-------|--------|-------|-------|--------|--------|---------------|
|        | PSNR↑ | SSIM↑ | LPIPS↓ | PSNR↑ | SSIM↑ | LPIPS↓ |        |               |
| D3DGS [63] | 22.54 | 0.715 | 0.215 | 21.54 | 0.675 | 0.287 | 42.6M | 10 mins |
| SoM [52] | 28.32 | 0.784 | 0.164 | 26.21 | 0.823 | 0.109 | 164.2M | 10 mins |
| D2RF [32] | 27.04 | 0.808 | 0.128 | 23.67 | 0.745 | 0.120 | 2.7M | 48 hrs |
| DyBluRF [47] | 26.24 | 0.788 | 0.159 | 24.53 | 0.864 | 0.079 | 1.3M | 48 hrs |
| De4DGS [58] | 28.49 | 0.791 | 0.154 | 26.62 | 0.871 | 0.059 | 754.6M | 20 hrs |
| Ours | **29.39** | **0.859** | **0.078** | **27.01** | **0.876** | **0.056** | 192.2M | 1 hr |

**Smoothing loss.** To improve scene motion representation, we introduce a smoothing constraint $\mathcal{L}_{smo}$ for the dynamic Gaussians' deformation. Similar to Shape-of-Motion [52], $\mathcal{L}_{smo}$ includes two components: a smoothness constraint on the $\mathbb{SE}(3)$ motion bases across adjacent frames and a smoothness constraint on the mean of dynamic Gaussians.

The overall training objective for the network is:

$$\mathcal{L} = \mathcal{L}_{rec} + \mathcal{L}_{geo} + \mathcal{L}_{smo} + \mathcal{L}_{spa}. \tag{13}$$

## 4 Experiments

**Evaluation datasets.** We evaluate our method on two datasets, including the one from D2RF [32] of defocus blur and the other one from DyBluRF [47] of motion blur. The first dataset from D2RF consists of 8 dynamic scenes, where each scene contains sharp stereo image sequences and their corresponding blurry images. The other one from DyBluRF contains 6 motion-blurred scenes composed of blurry stereo images and the corresponding sharp images. We train and evaluate our method on all sequences from the two datasets, using their left-view blurred sequences for training and the corresponding right-view sharp sequences for evaluation. Note that, similar to previous methods, the downsampled images in the two datasets are utilized for training and evaluation.

**Metrics.** Following previous work [52, 58, 32, 47], we quantitatively evaluate our performance on novel view synthesis using PSNR, SSIM [56], and LPIPS [67].

**Implementation Details.** We set the number of motion bases $N_b$ to 20, and the blur kernel size $K$ to 9. We employ the Adam optimizer [17] to jointly optimize the Gaussians and $\mathbb{SE}(3)$ motion bases and the BP-Net. The learning rates are set to $1.6 \times 10^{-4}$ for motion bases, and $5 \times 10^{-4}$ for

Table 2: **Quantitative comparison of novel view synthesis on the D2RF defocus blur dataset [32] and the DyBluRF motion blur dataset [47].** Note, BSSTNet [66] is a state-of-the-art video deblurring method, and "[66] + D3DGS [63]" means feeding the dynamic scene reconstruction [63] with deblurred images produced by [66].

| Method | Defocus Blur | | | Motion Blur | | |
|---|---|---|---|---|---|---|
| | PSNR ↑ | SSIM ↑ | LPIPS ↓ | PSNR ↑ | SSIM ↑ | LPIPS ↓ |
| D3DGS [63] | 22.54 | 0.715 | 0.215 | 21.54 | 0.675 | 0.287 |
| [66] + D3DGS [63] | 24.42 | 0.723 | 0.179 | 21.72 | 0.653 | 0.279 |
| SoM [52] | 28.32 | 0.784 | 0.164 | 26.21 | 0.823 | 0.109 |
| [66] + SoM [52] | 28.56 | 0.786 | 0.164 | 26.33 | 0.825 | 0.105 |
| Ours | **29.39** | **0.859** | **0.078** | **27.01** | **0.876** | **0.056** |

Table 3: **Quantitative ablation study on the D2RF defocus blur dataset [32] and the DyBluRF motion blur dataset [47].** Note, "w/o Unseen." refers to no unseen view information utilized.

| Method | Defocus Blur | | | Motion Blur | | |
|---|---|---|---|---|---|---|
| | PSNR ↑ | SSIM ↑ | LPIPS ↓ | PSNR ↑ | SSIM ↑ | LPIPS ↓ |
| w/o $\mathcal{L}_{spa}$ | 29.03 | 0.842 | 0.086 | 26.63 | 0.854 | 0.072 |
| w/o Shortcut | 29.12 | 0.845 | 0.086 | 26.74 | 0.852 | 0.078 |
| w/o DGD | 29.19 | 0.843 | 0.085 | 26.53 | 0.847 | 0.109 |
| w/o Unseen. | 29.09 | 0.836 | 0.090 | 26.66 | 0.853 | 0.075 |
| Full method | **29.39** | **0.859** | **0.078** | **27.01** | **0.876** | **0.056** |

BP-Net. The learning rate for the Gaussians is consistent with that of the original 3DGS [15]. We train each scene for 40,000 iterations and introduce unseen view information to constrain the scene starting from iteration 3,000. We set the iteration interval $N_u$ for using unseen view information to 5. Dynamic Gaussians densification is performed at $N_d = 2,500$ iterations. We introduce the unified blur synthesis model at iteration 3,500 and incorporate the blur-aware sparsity constraint at $N_{spa} = 5,500$ iterations. Training on a sequence of $512 \times 288$ resolution takes approximately 1 hour on an NVIDIA RTX 3090 GPU, with a rendering speed of 65.143 fps for the same resolution.

## 4.1 Comparison with State-of-the-Art Methods

**Compared methods.** We compare our method with various state-of-the-art methods including, DeformableGS [63], Shape-of-Motion [52], D2RF [32], DyBluRF [47], and Deblur4DGS [58], where the method of D2RF [32] is designed for defocus blur, while the methods of DyBluRF [47] and Deblur4DGS [58] are tailored for motion blur. For fair comparison, we produce their results using publicly-available implementation or trained models provided by the authors with the recommended parameter setting.

**Comparison on defocus blur.** Tables 1 and 2 report the quantitative results on the defocus blur dataset, where we can see that our method outperforms other methods, even existing methods that are fed with deblurred images produced by a state-of-the-art video deblurring method. The reason is that video deblurring methods cannot effectively ensure 3D scene consistency in the deblurred images. Figures 3 and 4 further present a visual comparison of novel-view synthesis. As shown, our method exhibits a clear advantage in producing high-quality novel views in both dynamic and static regions, while the compared methods struggle to maintain structural details and provide sharp results. Please see the supplementary material for additional comparison results, including both image and video results of defocus blur.

**Comparison on motion blur.** As shown in Tables 1 and 2, our method quantitatively outperforms the compared methods on all three metrics, manifesting its effectiveness in dealing with motion blurred monocular videos. We also show the visual comparison in Figure 5. As can be seen, our method produces results with more incomplete structure details and sharper appearance. Please see the supplementary material for additional comparison results, including both image and video results of motion blur.

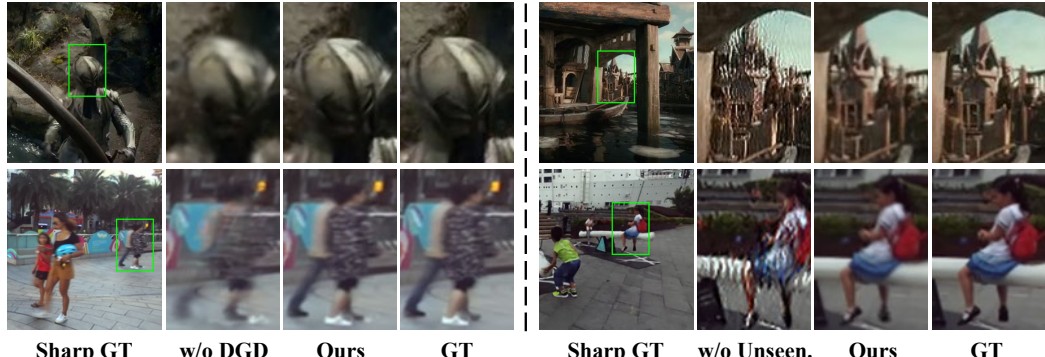

Figure 6: **Left:** Effect of dynamic Gaussian densification (DGD). **Right:** Effect of leveraging unseen view information. Note, "w/o Unseen." refers to no unseen view information utilized.

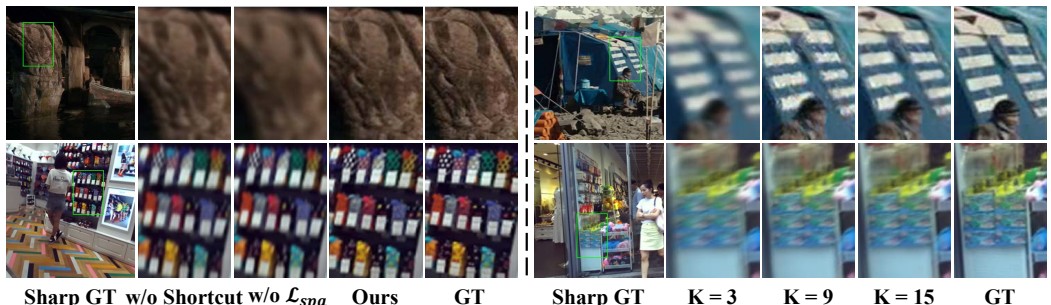

Figure 7: **Left:** Effect of blur-aware sparsity Constraint $\mathcal{L}_{spa}$ and the shortcut in BP-Net. **Right:** Effect of varying blur kernel size $K$.

### 4.2 More Analysis

**Ablation study.** We conduct an ablation study to evaluate the contribution of each component in our model. Specifically, we evaluate the effect of (i) removing blur-aware sparsity constraint (w/o $\mathcal{L}_{spa}$), (ii) removing the shortcut in BP-Net (w/o Shortcut), (iii) initializing the dynamic Gaussians using only the visible 2D tracking points in the canonical frame, without utilizing dynamic Gaussian densification (w/o DGD), (iv) removing unseen view information during training (w/o Unseen.). We report the quantitative results in Table 3, where we can see that each of our design has a clear contribution. In addition, we in Figures 6 and 7 also qualitatively validate the necessity of each component in our model. Besides, we also assess the influence of different blur kernel size in Figure 7. As shown, a larger blur kernel ($K$) helps produce better results, but this trend becomes less obvious when $K > 9$.

**Limitations.** Since our method relies on 2D image priors, errors arising from 2D prediction such as depth estimation and segmentation may degrade the overall performance of our method. Moreover, for dynamic scenes with large non-rigid motion blur, our method, as well as other state-of-the-art methods, would fail to produce high-quality novel-view results free of visual artifacts, as demonstrated in Figure 8. Finally, similar to the vanilla 3DGS, our method has to be optimized for each scene separately.

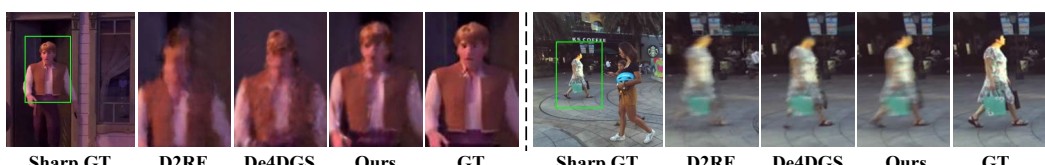

Figure 8: **Failure case.** Our method may fail to handle a dynamic scene with large non-rigid motion blur. Note, the left column demonstrates results for defocus blur, while the right column presents motion blur outcomes.

# 5 Conclusion

We have presented a unified framework for generating high-quality novel views from defocused and motion-blurred monocular videos. In contrast to previous methods, which are either tailored for defocus blur or motion blur, we propose to model both blur types using blur kernel-based convolution. To this end, we develop a blur prediction network exploiting blur-related scene and camera information to estimate reliable blur kernels and pixel-wise intensity. Besides, we introduce a dynamic Gaussian densification strategy to mitigate the lack of Gaussians for incomplete regions and boost the performance of novel view synthesis by incorporating unseen view information to constrain scene optimization. Extensive experiments demonstrate that our method outperforms the state-of-the-art methods in generating photorealistic novel view synthesis from defocused and motion-blurred monocular videos.

**Acknowledgement.** This work was supported by the National Natural Science Foundation of China (62471499), the Guangdong Basic and Applied Basic Research Foundation (2023A1515030002).

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

# A Implementation Details

**Network Architecture.** The detailed architecture of Scene Feature Extractor Network and Blur Prediction Network in our framework are illustrated in Figure 9. We enhance the ability to capture high-frequency details by using positional encoding $p$ for pixel coordinates $x$ and a discrete variable embedding module $e$ (implemented with PyTorch) for camera indices $i$.

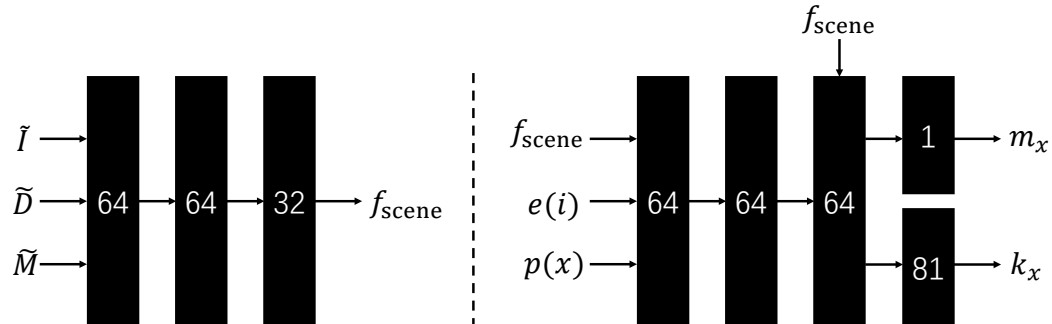

Figure 9: **Scene Feature Extractor Network (Left).** Scene Feature Extractor Network takes rendered image $\tilde{I}$, rendered depth $\tilde{D}$ and rendered mask $\tilde{M}$ as input, and outputs scene feature $f_{scene}$. Besides the last layer, each layer outputs 64-dimensional features with ReLU activations. **Blur Prediction Network (Right).** Blur Prediction Network takes scene feature $f_{scene}$, camera embedding vector $e(i)$ and pixel positional encoding $p(x)$ as input, and outputs blur kernel $k_x$ and blur intensity $m_x$. Each layer outputs 64-dimensional features with ReLU activations, except for the last layer. Note, in the last layer $m_x$ is obtained via Sigmoid activations, while $k_x$ is obtained via Softmax activations.

# B Additional Quantitative and Qualitative Results

We compare our method with dynamic scene reconstruction methods [63, 52] that use video-deblurred images as input. Figure 10 presents a visual comparison of novel view synthesis on the motion blur dataset, where we can see that our method outperforms existing methods that are fed with deblurred images produced by a state-of-the-art video deblurring method. The reason is that video deblurring methods cannot effectively ensure 3D scene consistency in the deblurred images. Please see the video supplementary material for additional novel view synthesis comparison results.

## B.1 Novel View Synthesis Comparison

**D2RF and DyBluRF Dataset.** We compare our approach against BAGS [41] and De3DGS [21], two methods designed to reconstruct sharp static scenes from blurred static images, and evaluate them on the D2RF [32] and DyBluRF [47] datasets. Table 4 and Figure 11 present the comparison results. Clearly, our method demonstrates significant advantages over other methods, producing sharper novel view images while better preserving realistic motion details.

**D2RF-v2 and DyBluRF-v2 Dataset.** We evaluate our method on two datasets (DyBluRF-v2 and D2RF-v2) with both motion and defocus blur occurring simultaneously. Note that we obtain the DyBluRF-v2 dataset by applying depth-of-field (DoF) rendering technique in Bokehme [42] to the original DyBluRF dataset [47] to simulate defocus blur, and create the D2RF-v2 dataset with motion blur by processing the original D2RF dataset [32] using the motion blur generation method in Davanet [70]. Table 5 present the comparison results. As shown, our method clearly outperforms all the compared methods on the two datasets, verifying the advantage of our method in handling cases with motion and defocus blur occurring jointly.

**Deblur-NeRF Dataset.** We compare our approach against De3DGS [21], which is designed to reconstruct sharp static scenes from blurred static images, and evaluate them on the Deblur-NeRF [34] dataset. Table 6 and Figure 12 present the comparison results. Clearly, our method demonstrates significant advantages over other methods, producing sharper novel view images.

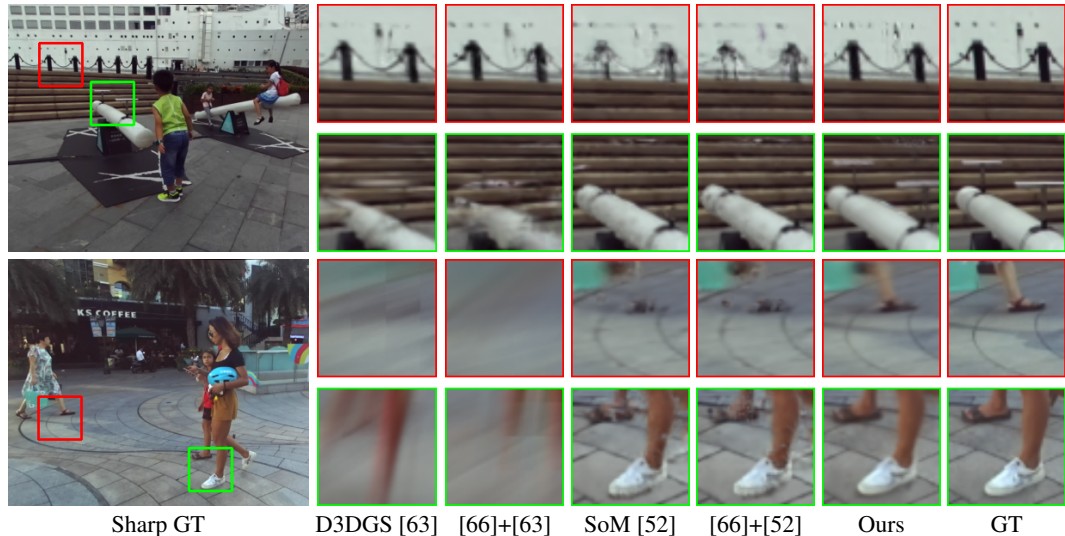

| Sharp GT | D3DGS [63] | [66]+[63] | SoM [52] | [66]+[52] | Ours | GT |

Figure 10: **Visual comparison of novel view synthesis on the DyBluRF motion blur dataset [47].** Here, we also compare with methods fed with deblurred images produced by a state-of-the-art video deblurring method [66] to manifest the effectiveness of our method.

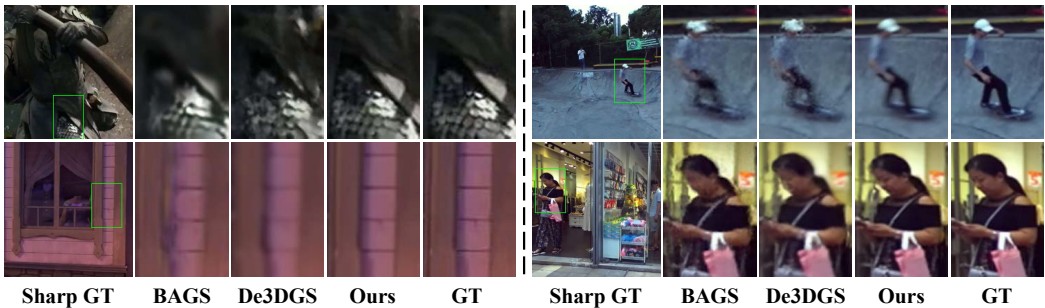

| **Sharp GT** | **BAGS** | **De3DGS** | **Ours** | **GT** | **Sharp GT** | **BAGS** | **De3DGS** | **Ours** | **GT** |

Figure 11: **Visual comparison of novel view synthesis.** Here, we compare our method with methods that are designed to reconstruct sharp static scenes from blurred static scene images. Note, the left column demonstrates results for defocus blur, while the right column presents motion blur outcomes.

Table 4: **Quantitative comparison of novel view synthesis on the D2RF defocus blur dataset [32] and the DyBluRF motion blur dataset [47].**

| Method | Defocus Blur | | | Motion Blur | | |
|---|---|---|---|---|---|---|
| | PSNR ↑ | SSIM ↑ | LPIPS ↓ | PSNR ↑ | SSIM ↑ | LPIPS ↓ |
| BAGS [41] | 24.41 | 0.730 | 0.167 | 24.27 | 0.723 | 0.208 |
| De3DGS [21] | 23.74 | 0.716 | 0.190 | 22.45 | 0.689 | 0.253 |
| Ours | **29.39** | **0.859** | **0.078** | **27.01** | **0.876** | **0.056** |

## B.2 Deblurring Comparison

We compare the deblur ring capability of our method with a broad range of existing methods, including 3DGS- and NeRF-based methods for both dynamic and static scenes [32, 58, 41, 21], as well as transformer-based video deblurring method [66]. Specifically, we compare the sharp images produced at training views. For 3DGS- and NeRF-based methods, these images are rendered from the trained sharp scene representations using the same training views. Table 7 and Figure 13 present the comparison results. Our method outperforms 3DGS- and NeRF-based deblurring approaches and achieves performance comparable to state-of-the-art video deblurring methods.

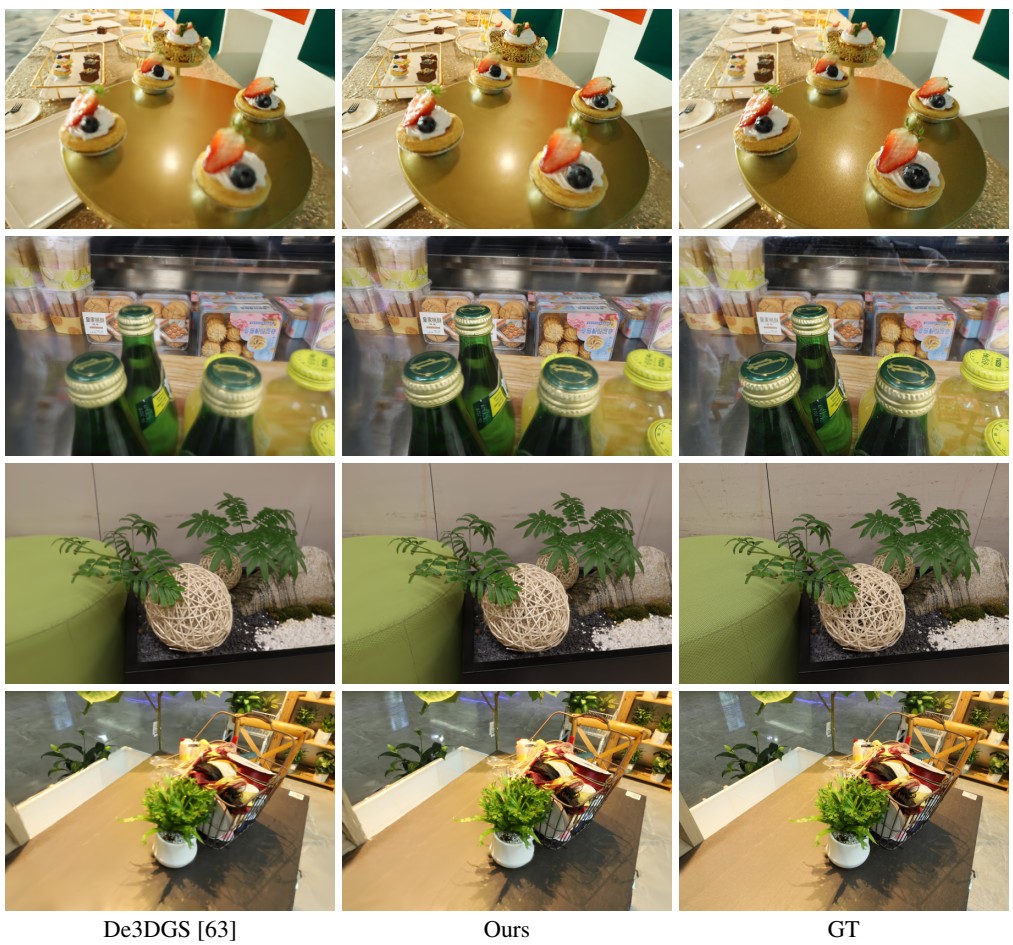

|   De3DGS [63]   |   Ours   |   GT   |

Figure 12: **Visual comparison of novel view synthesis on the Deblur-NeRF dataset [34].**

Table 5: **Quantitative comparison of novel view synthesis on the D2RF-v2 defocus blur dataset and the DyBluRF-v2 motion blur dataset.** The numerical results of defocus blur are obtained on the Shop and Car scenes of the D2RF-v2 dataset, and the numerical results of motion blur are obtained on the Man and Seesaw scenes of the DyBluRF-v2 dataset.

| Method | Defocus Blur | | | Motion Blur | | |
|---|---|---|---|---|---|---|
| | PSNR↑ | SSIM↑ | LPIPS↓ | PSNR↑ | SSIM↑ | LPIPS↓ |
| D3DGS [63] | 23.66 | 0.739 | 0.257 | 21.75 | 0.655 | 0.289 |
| SoM [52] | 29.04 | 0.820 | 0.094 | 27.28 | 0.791 | 0.103 |
| D2RF [32] | 27.82 | 0.795 | 0.132 | 25.89 | 0.722 | 0.133 |
| DyBluRF [47] | 27.30 | 0.771 | 0.150 | 26.54 | 0.753 | 0.112 |
| De4DGS [58] | 29.74 | 0.856 | 0.078 | 27.97 | 0.824 | 0.087 |
| Ours | **30.26** | **0.885** | **0.062** | **28.55** | **0.859** | **0.064** |

Table 6: **Quantitative comparison of novel view synthesis on the Deblur-NeRF dataset [34].**

| Method | Defocus Blur | | | Motion Blur | | |
|---|---|---|---|---|---|---|
| | PSNR ↑ | SSIM ↑ | LPIPS ↓ | PSNR ↑ | SSIM ↑ | LPIPS ↓ |
| De3DGS [21] | 23.71 | 0.747 | 0.110 | 26.61 | 0.822 | 0.108 |
| Ours | **24.22** | **0.768** | **0.095** | **27.14** | **0.835** | **0.096** |

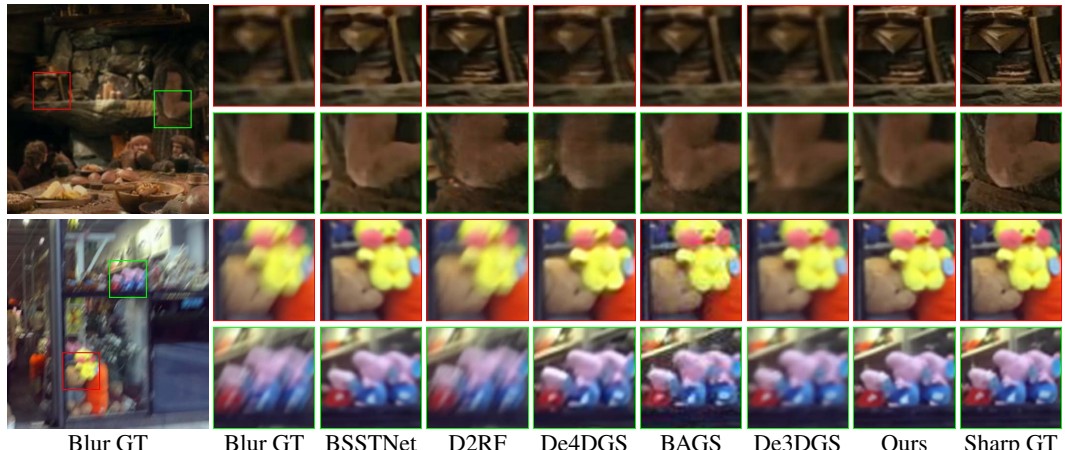

| Blur GT | Blur GT | BSSTNet | D2RF | De4DGS | BAGS | De3DGS | Ours | Sharp GT |

Figure 13: **Visual comparison of deblurring.** Our method enables the synthesis of high-quality deblurring results for videos with defocus blur (top) and motion blur (bottom).

Table 7: **Quantitative comparison of deblurring on the D2RF defocus blur dataset [32] and the DyBluRF motion blur dataset [47].**

| Method | Defocus Blur | | | Motion Blur | | |
|---|---|---|---|---|---|---|
| | PSNR ↑ | SSIM ↑ | LPIPS ↓ | PSNR ↑ | SSIM ↑ | LPIPS ↓ |
| BSSTNet [66] | 33.54 | 0.961 | 0.039 | **33.71** | **0.965** | **0.030** |
| D2RF [32] | 34.33 | 0.976 | 0.028 | 32.14 | 0.949 | 0.049 |
| De4DGS [58] | 32.92 | 0.953 | 0.044 | 33.27 | 0.958 | 0.035 |
| BAGS [41] | 30.69 | 0.940 | 0.084 | 30.55 | 0.935 | 0.092 |
| De3DGS [21] | 30.36 | 0.941 | 0.090 | 29.84 | 0.924 | 0.105 |
| Ours | **34.85** | **0.977** | **0.027** | 33.45 | 0.960 | 0.036 |

## C  Additional Ablation Results

### C.1  Ablation on BP-Net

We conduct an ablation study to evaluate the contribution of BP-Net. Specifically, we compare three different blur modeling methods: (i) the motion blur and defocus blur modeling method used in De3DGS [21] (w/ blur modeling in De3DGS [21]), (ii) the motion blur modeling method in De4DGS [58] (w/ blur modeling in Deblur4DGS [58]), and (iii) the defocus blur modeling method in D2RF [32] (w/ blur modeling in D2RF [32]). We report the quantitative results in Table 8, where we can see that our method with the proposed BP-Net produces better results than these alternatives, demonstrating the effectiveness of the BP-Net.

Table 8: **Effect of BP-Net.**

| Method | Defocus Blur | | | Motion Blur | | |
|---|---|---|---|---|---|---|
| | PSNR↑ | SSIM↑ | LPIPS↓ | PSNR↑ | SSIM↑ | LPIPS↓ |
| w/ blur modeling in De3DGS [21] | 28.31 | 0.812 | 0.098 | 26.05 | 0.823 | 0.118 |
| w/ blur modeling in De4DGS [58] | 28.63 | 0.829 | 0.094 | 26.74 | 0.859 | 0.060 |
| w/ blur modeling in D2RF [32] | 28.96 | 0.832 | 0.094 | 26.30 | 0.825 | 0.109 |
| Ours with BP-Net | **29.39** | **0.859** | **0.078** | **27.01** | **0.876** | **0.056** |

### C.2  Ablation on Blur Kernel Size

Table 9 further perform quantitative evaluation on how different blur kernel sizes (denoted as $K$) affect the performance of our method. As shown, a larger blur kernel helps to obtain better results. However, this trend becomes less obvious when $K$ is larger than 9. To balance the performance and the computational cost, we thus choose $K = 9$ as our default choice.

Table 9: **Effect of varying blur kernel size** $K$. The numerical results of defocus blur are obtained on the Gate and Dock scenes of the D2RF dataset, and the numerical results of motion blur are obtained on the Skating and Man scenes of the DyBluRF dataset.

| Blur kernel size | Defocus Blur | | | Motion Blur | | |
| --- | --- | --- | --- | --- | --- | --- |
| | PSNR↑ | SSIM↑ | LPIPS↓ | PSNR↑ | SSIM↑ | LPIPS↓ |
| $K$=5 | 27.84 | 0.817 | 0.098 | 29.53 | 0.906 | 0.092 |
| $K$=7 | 28.12 | 0.836 | 0.074 | 29.79 | 0.913 | 0.067 |
| $K$=9 | 28.29 | 0.842 | 0.067 | 30.01 | 0.921 | 0.052 |
| $K$=11 | 28.30 | 0.843 | 0.066 | 30.02 | 0.920 | 0.050 |
| $K$=13 | 28.31 | 0.842 | 0.067 | 30.04 | 0.922 | 0.051 |

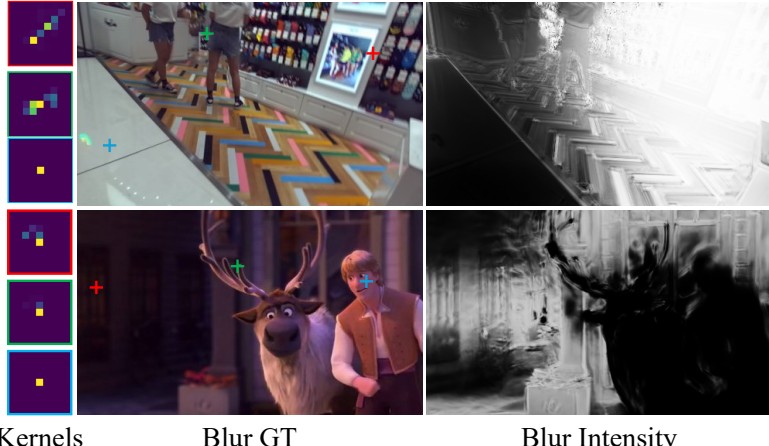

Kernels     Blur GT        Blur Intensity

Figure 14: **Visualization of the blur kernel and blur intensity predicted by BP-Net.** Note that the top row shows an image with defocus blur, while the bottom row shows an image with motion blur. In the blur ground truth (GT) image, blue markers indicate pixels with almost no blur, green markers denote pixels with mild blur, and red markers represent pixels with severe blur. A higher blur intensity value corresponds to a more heavily blurred area. Clearly, BP-Net can accurately predict blur regions in images with different types of blur and estimate the corresponding blur kernels at pixel locations with varying blur levels.

# D    More Analysis on BP-Net

Our method represents image blurring effects using two components: a blur kernel $k$ and a blur intensity $m$, as illustrated in Figure 14. The blur intensity $m$ effectively emphasizes the spatial regions affected by blur within each training image. Moreover, the type of blur can be intuitively inferred from the estimated kernels. Specifically, kernels corresponding to motion blur capture structured trajectories that reflect the camera's movement, while those associated with defocus blur present Gaussian-shaped patterns that vary with the distance of the pixel from the focal plane.

To evaluate the accuracy of the blur kernel $k$ predicted by BP-Net, we compare the ground truth blur kernel with the blur kernel predicted by BP-Net on a blurry monocular video dataset with ground truth blur kernel. To this end, we construct two datasets with ground truth blur kernels, referred to as D2RF-v3 and DyBluRF-v3, by randomly sampling two global Gaussian and linear distribution blur kernels of size 9 x 9 and then respectively applying them to the ground truth sharp images in D2RF [32] and DyBluRF [47] to obtain the corresponding blurry images. With the two datasets, we quantitatively compare our estimated blur kernels and the ground truth blur kernels using PSNR and KL divergence as metrics. Table 10 and Figure 15 present the comparison results. Clearly, our estimated blur kernels are highly similar to the ground truth blur kernels in numerical metrics, manifesting the effectiveness of the BP-Net in predicting different types of blur.

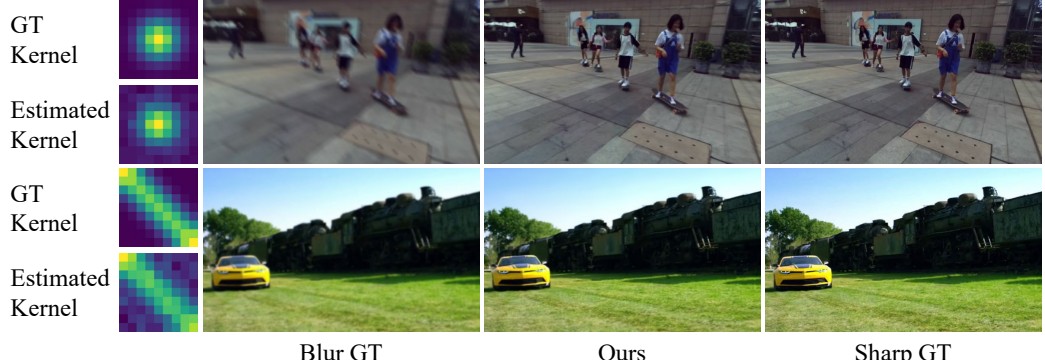

| | Blur GT | Ours | Sharp GT |

Figure 15: **Visual comparison of estimated kernel on the D2RF-v3 defocus blur dataset and the DyBluRF-v3 motion blur dataset.** Note that the top row shows an image with defocus blur, while the bottom row shows an image with motion blur. Clearly, BP-Net can accurately estimate blur kernels from various types of blurry images and thus recover sharp images.

Table 10: **Comparison of ground truth kernels and estimated kernels.**

| | D2RF-v3 | DyBluRF-vs |
|---|---|---|
| PSNR↑ | 32.516 | 29.941 |
| KL Div.↓ | 0.214 | 0.247 |

## E   Robustness to Preprocessing and Segmentation Errors

In the Table 11, we quantitatively evaluate how errors from external preprocessing steps affect the robustness of our method. To simulate errors from depth estimation, we randomly scale and shift the estimated depth maps within the range of [0.8, 1.2] and [-20, 20], respectively. To simulate errors from 2D point tracking and SAM, we randomly shift the 2D tracking points within the range of [-30, 30], and randomly add or delete five 25 X 25 mask regions towards the mask predicted by SAM. As shown, our results produced with the artificially perturbed depth, tracking points, and motion mask are comparable to those produced with the originally estimated depth, tracking points, and motion mask, indicating that our method has some tolerance to errors from external preprocessing steps.

Table 11: **Analysis on the impact of errors from external preprocessing steps.**

| Method | Defocus Blur | | | Motion Blur | | |
|---|---|---|---|---|---|---|
| | PSNR↑ | SSIM↑ | LPIPS↓ | PSNR↑ | SSIM↑ | LPIPS↓ |
| Ours w/ depth perturbation | 29.19 | 0.844 | 0.088 | 26.79 | 0.862 | 0.074 |
| Ours w/ tracking perturbation | 29.21 | 0.854 | 0.084 | 26.86 | 0.874 | 0.060 |
| Ours w/ mask perturbation | 29.25 | 0.854 | 0.085 | 26.74 | 0.865 | 0.067 |
| Ours | 29.39 | 0.859 | 0.078 | 27.01 | 0.876 | 0.056 |

