# OpenReview forum: "Dynamic Gaussian Splatting from Defocused  and Motion-blurred Monocular Videos"
_NeurIPS.cc/2025/Conference — NeurIPS 2025 poster_

### Official Review · Reviewer_Uuup · 2025-06-15

**Clarity:** 3
**Significance:** 3
**Originality:** 3
**Rating:** 5
**Confidence:** 3

**Summary:**

The paper proposes a method that jointly models defocus and motion blur using a neural network that predicts per-pixel blur kernels and intensities. This unified blur modeling guides the optimization of dynamic and static Gaussians for sharper reconstruction from blurry monocular videos.

**Questions:**

1. Could the authors clarify how adding an intensity output to the blur kernel network alone prevents it from deforming jointly with the 3DGS representation?

2. Why not consider using a pre-trained large model to simulate blur instead of training a network together with 3DGS, which might avoid the deformation issue altogether?

3. A straightforward idea would be to first apply a network to generate the blurred image, then feed the clear image to 3DGS for reconstruction. Could the authors elaborate on the main advantages of the proposed coupled approach compared to such a decoupled pipeline?

If the authors can provide clear and convincing answers to these questions, I am willing to consider increasing my evaluation score.

**Ethical Concerns:**

["NO or VERY MINOR ethics concerns only"]

**Final Justification:**

My initial concerns about novelty and justification have been addressed, and I’ve raised my score.

Key reasons include:

1. The authors convincingly argue that joint optimization is better than a simple two-stage pipeline, as it preserves critical information.

2. Their choice to avoid large models is well-justified

**Limitations:**

yes

**Paper Formatting Concerns:**

No major formatting issues noticed

**Quality:**

3

**Strengths And Weaknesses:**

**Strengths:**
1. The paper is very well written, with a clear and progressive structure that makes it easy to follow.
2. The experiments are reasonably comprehensive, including both main results and ablation studies.
3. The work focuses sharply on the core challenge of jointly handling motion blur and defocus blur, which is timely and relevant.

**Weaknesses:**
1. The interaction between different modules could be stronger — the current design resembles a standard 3DGS pipeline followed by a separate blur modeling network, rather than a deeply integrated solution.
2. Some methodological details are under-explained. For instance, while adding an intensity output to the blur kernel network is a sensible idea, it is unclear how this modification alone prevents the network from deforming jointly with 3DGS — since it remains using a network to generate blur.

---

> ### Author Rebuttal · Authors · 2025-07-31
>
> **_(Q1) The current design resembles a standard 3DGS pipeline followed by a separate blur modeling network, rather than a deeply integrated solution._**
>
> Indeed, our framework is not highly integrated in architecture. This is mainly because we adopt kernel-based post-processing to achieve unified modeling of both motion and defocus blur. Though our method is not a deeply integrated solution, it still outperforms previous methods, demonstrating its effectiveness.
>
> **_(Q2) Clarify how adding an intensity output to the blur kernel network alone prevents it from deforming jointly with the 3DGS representation._**
>
> By adopting an intensity output, we are able to identify the blurry and sharp regions in the blurry input image, with which we can enforce the blur kernel network to focus on the blurry regions and also use the sharp regions to constrain the Gaussian rendered image, thereby avoiding deforming jointly with the 3DGS representation.
>
> **_(Q3) Why not consider using a pre-trained large model to simulate blur instead of training a network together with 3DGS, which might avoid the deformation issue altogether?_**
>
> Thank you for this inspiring comment. We agree that this is a promising direction to avoid the deformation issue. However, existing pre-trained large models basically recover sharp images from blurry inputs in an end-to-end manner without explicitly modeling blur. Although we have tried our best to fine-tune some pre-trained models (Restormer [1] and DiffBIR [2]) to estimate the blur kernel, we failed to obtain satisfactory results within the rebuttal period. We will keep working on this direction in our future work.
>
> **_(Q4) A straightforward idea would be to first apply a network to generate the blurred image, then feed the clear image to 3DGS for reconstruction. Could the authors elaborate on the main advantages of the proposed coupled approach compared to such a decoupled pipeline?_**
>
> The main advantage of our approach is that our method can obtain spatiotemporally consistent scene representation (see Table 2 and Figure 4 in the paper, as well as Figure 2 in the supplementary material for comparison), which is hardly achieved by the decoupled pipeline. The reason is that the decoupled pipeline performs deblurring in 2D image space without considering the 3D information of the scene, and thus there usually exists noticeable spatiotemporal inconsistencies between the deblurred frames, which in turn cause spatiotemporal inconsistency in 3DGS reconstructed scene.
>
> **_References_**
>
> [1] Zamir, S. W., Arora, A., Khan, S., Hayat, M., Khan, F. S., & Yang, M. H. Restormer: Efficient Transformer for High-Resolution Image Restoration. CVPR 2022.
>
> [2] Lin, X., He, J., Chen, Z., Lyu, Z., Dai, B., Yu, F., ... & Dong, C. DiffBIR: Towards Blind Image Restoration with Generative Diffusion Prior. ECCV 2024.

---

> > ### Author Response · Authors · 2025-08-03
> >
> > Dear Reviewer Uuup,
> >
> > Thank you again for your time and valuable feedback on our paper. We've carefully addressed all your concerns in our rebuttal and hope these clarifications sufficiently address them. If you have any follow-up questions or require additional details, we are very happy to discuss further.
> >
> > Best regards,
> >
> > The authors

---

> > > ### Comment · Reviewer_Uuup · 2025-08-04
> > > **Score updated after rebuttal**
> > >
> > > Dear Authors,
> > >
> > > Thank you for the detailed clarifications. I appreciate the thoughtful responses and have updated my score accordingly.
> > >
> > > Best regards,
> > > Reviewer Uuup

---

### Official Review · Reviewer_qGyt · 2025-06-30

**Clarity:** 2
**Significance:** 2
**Originality:** 2
**Rating:** 4
**Confidence:** 4

**Summary:**

This paper proposes a unified 3D dynamic Gaussian framework for novel view synthesis from monocular blurry videos, capable of simultaneously handling motion blur and defocus blur. The framework employs a blur kernel prediction network to predict per-pixel blur kernels, which are regularized through blur-aware sparsity constraints. Additionally, the framework introduces a dynamic Gaussian densification strategy to optimize unseen views.

**Questions:**

1. The methodology section presents a disorganized description of the overall framework, lacking clear explanations of the relationships between different modules. For example, Line 82, R20 does not clearly explain what the number 20 specifically represents; Formula 2 does not provide a clear explanation of how the dynamic Gaussian achieves scaling transformation; the specific meaning in Formula 3 is not explained. The description of the two modules in Figure 2 is insufficiently clear, such as how the Image, Depth, and Mask in the right figure are obtained, and how the Blur-aware Sparsity Constrain performs backpropagation.

2. The description of the blur kernel estimation network is unclear: No explicit justification is provided for the chosen blur kernel size. The physical meaning of "blur intensity mx" is not defined. The paper fails to explain why blur-aware sparsity constraints can stabilize mx.

3. Regarding the training strategy: How is "restrict the selection to unseen views close to the training views" implemented?

4. Experimental setup issues: Key hyperparameter values are not specified. The dataset used was not proposed by Deblur4DGS (incorrect citation).

5. Line 3: "diference" is misspelled (should be "difference").

6. Figure 2: We recommend using dashed lines to represent the Gradient Flow for better visual distinction.

**Ethical Concerns:**

["NO or VERY MINOR ethics concerns only"]

**Final Justification:**

The authors have addressed my concerns, and I have updated the rating.

**Limitations:**

yes

**Quality:**

2

**Strengths And Weaknesses:**

Strengths: This paper presents an extension of blur kernel estimation from static scene novel view synthesis to the more challenging monocular video setting, proposing a unified framework that simultaneously addresses both motion and defocus blur.

Weaknesses: The technical presentation suffers from significant clarity issues - the methodological framework lacks coherent explanation, with ambiguous descriptions and unclear mathematical derivations throughout the paper. Furthermore, the experimental section remains inadequately detailed, particularly regarding crucial implementation specifics like hyperparameter configurations, substantially limiting reproducibility and evaluation rigor. See 'questions' for details.

---

> ### Author Rebuttal · Authors · 2025-07-31
>
> **_(Q1) Line 82, what the number 20 specifically represents in R20._**
>
> The number 20 denotes the total number of SE(3) motion bases. We will clarify this in the paper.
>
> **_(Q2) How the dynamic Gaussian achieves scaling transformation in Formula 2._**
>
> Note, we do not perform scaling transformation of dynamic Gaussian in Formula 2, as we aim to achieve dynamic Gaussian deformation in Formula 2. Therefore, we only adjust the pose parameters (position and rotation).
>
> **_(Q3) The meaning in Formula 3 is not explained._**
>
> The goal of Formula 3 is to obtain the center of the densified dynamic Gaussian by performing depth reprojection of each pixel in a frame to its corresponding 3D point location.
>
> **_(Q4) How the Image, Depth, and Mask in the right part of Figure 2 obtained, and how the Blur-aware Sparsity Constrain performs backpropagation._**
>
> The mentioned image, depth, and mask are obtained by rasterizing both dynamic and static Gaussians. During training, the gradient of blur-aware sparsity constraint flows to BP-Net through the learnable blur kernel, thereby allowing backpropagation.
>
> **_(Q5) No explicit justification is provided for the chosen blur kernel size._**
>
> In addition to the visual justification in Figure 7, we in the table below further perform quantitative evaluation on how different blur kernel sizes (denoted as K) affect the performance of our method. As shown, a larger blur kernel helps to obtain better results. However, this trend becomes less obvious when K is larger than 9. To balance the performance and the computational cost, we thus choose K=9 as our default choice.  We will add the table to the supplementary material.
>
> |Blur kernel size||Defocus blur (D2RF [1])||||Motion Blur (DyBluRF [2])||
> |:------------:|:---:|:----------:|:---:|:---:|:---------:|:---:|:---:|
> ||PSNR↑|SSIM↑|LPIPS↓||PSNR↑|SSIM↑|LPIPS↓|
> |K=5|27.84|0.817|0.098||29.53|0.906|0.092|
> |K=7|28.12|0.836|0.074||29.79|0.913|0.067|
> |K=9|28.29|0.842|0.067||30.01|0.921|0.052|
> |K=11|28.30|0.843|0.066||30.02|0.920|0.050|
> |K=13|28.31|0.842|0.067||30.04|0.922|0.051|
>
> **_(Q6)  The physical meaning of "blur intensity mx" is not defined._**
>
> The physical meaning of $m_x$ is the degree of blur of a pixel $x$, and a larger $m_x$ indicates a blurrier pixel.
>
> **_(Q7) The paper fails to explain why blur-aware sparsity constraints can stabilize mx._**
>
> Thank you for careful review. In fact, what we intend to express in Line 162-163 is that we adopt the blur-aware sparsity constraint $L_{spa}$ for training only when the blur intensity $m_x$ has been trained for a number of $N_{spa}$ iterations to be stable, rather than claiming that the constraint can stabilize $m_x$. We will revise the paper to clarify this.
>
> **_(Q8) How is "restrict the selection to unseen views close to the training views" implemented?_**
>
> We implement this by generating two types of unseen views: (i) parallel-unseen views: generated by interpolating between adjacent training views along the camera trajectory, (ii) perpendicular-unseen views: generated by first computing a local perpendicular direction to the camera trajectory and then perturbing the training view’s camera center along this perpendicular direction by a distance of [0.5, 1] (normalized units). We will clarify this in the paper.
>
> **_(Q9) Key hyperparameter values are not specified._**
>
> In fact, we give all implementation details in the supplementary material, including the hyperparameter values. We will move the implementation details to the paper.
>
> **_(Q10) Incorrect citation of the dataset._**
>
> Thank you for careful checking. We will address the issue in the revised paper.
>
> **_(Q11) Line 3: "diference" is misspelled (should be "difference")._**
>
> We will correct this typo.
>
> **_(Q12) Using dashed lines in Figure 2 to represent the Gradient Flow._**
>
> We will update Figure 2 according to your suggestion.
>
> **_References_**
>
> [1] Luo, X., Sun, H., Peng, J., & Cao, Z. Dynamic neural radiance field from defocused monocular video. ECCV 2024.
>
> [2] Sun, H., Li, X., Shen, L., Ye, X., Xian, K., & Cao, Z. Dyblurf: Dynamic neural radiance fields from blurry monocular video. CVPR 2024.

---

> > ### Author Response · Authors · 2025-08-03
> >
> > Dear Reviewer qGyt,
> >
> > Thank you again for your time and valuable feedback on our paper. We've carefully addressed all your concerns in our rebuttal and hope these clarifications sufficiently address them. If you have any follow-up questions or require additional details, we are very happy to discuss further.
> >
> > Best regards,
> >
> > The authors

---

> > ### Comment · Reviewer_qGyt · 2025-08-06
> > **Thanks for the rebuttal**
> >
> > Thank you for your detailed response. The authors have addressed some of my concerns regarding the formula parameters and experimentally validating the impact of different blur kernel sizes on the results. However, I still have reservations. Since Formula 2 does not perform scaling transformations and the proposed method only incorporates translation and rotation parameters, I wonder how it handles scenarios involving object resizing with dynamic gaussian.

---

> > > ### Author Response · Authors · 2025-08-06
> > >
> > > Dear Reviewer qGyt,
> > >
> > > Thank you for the feedback and the insightful question. Although we do not perform scaling transformations in Formula 2, our method can also handle scenarios with object resizing. The reason is that our dynamic Gaussian densification is designed to add small-scale dynamic Gaussians to dynamic regions of each video frame, which helps ensure that there always exists a certain number of dynamic Gaussians for representing object with varying sizes. We will provide video results in the supplementary material to demonstrate the ability of our method in dealing with scenarios with object resizing.
> > >
> > > Best regards,
> > >
> > > The authors

---

> ### Comment · Area_Chair_WFEn · 2025-08-05
>
> Dear reviewer qGyt,
>
> Please be reminded that you have to finish the Mandatory Acknowledgement after reading the rebuttals. Please also kindly provide additional feedbacks on whether your concerns are addressed by the rebuttals.
>
> Thanks, AC

---

### Official Review · Reviewer_ocL4 · 2025-07-02

**Clarity:** 2
**Significance:** 3
**Originality:** 2
**Rating:** 4
**Confidence:** 4

**Summary:**

The paper addresses the challenge of building dynamic Gaussian splatting representations from monocular videos degraded by two types of blurs: motion blurs and defocus blurs. While previous methods focus solely on one type of blur, the authors present a framework that handles problems from both types of blurs simultaneously. This is achieved by a blur prediction network (BP-Net) for synthesizing blurs and a densification strategy for dynamic Gaussian splatting, which fills in the missing parts between dynamic Gaussians. In the experiment section, the paper shows outperforming metrics in both benchmarks for separate blur types.

**Questions:**

Regarding the Efficacy of the Proposal
1. How well is BP-Net trained for different types of blurs? Comparing generated and estimated kernels can clarify the importance of the suggested BP-Net in this framework.
2. Why did the authors ablate on the shortcut connections of the BP-Net in Table 3, rather than the BP-Net itself, if the authors wanted to demonstrate the BP-Net itself as their contributions, as in Line 43? Either more clearly stating the exact contributions or restructuring the ablation studies section to better align with the contributions will help.
3. What if the two types of blurs occur jointly? Can the proposed method also handle this type of degradation?

Regarding Presentation
1. Please clarify which part is the novel contribution of the authors in section 3. As shown in Figure 2, the overall pipeline involves several pre-built solutions, including SAM, Depth-Anything, and TAPIR. Therefore, it seems crucial to distinguish the paper’s own contributions from those in the manuscript. This version, for example, in lines 76-113, mixes the off-the-shelf and the developed methods in the text. It is challenging to identify which parts constitute the novel contributions.
2. How frequently does the densification take place (What is N_d)? Showing exact figures for the hyperparameters can boost the clarity of the manuscript. For example, in Line 85 and Line 103, the number of motion bases is only denoted as an abstract symbol N_d. In Line 171, we have N_u, whose value is also not specified. It will not hurt the generality of the method to provide a rough scale of the variables used in the method.

**Ethical Concerns:**

["NO or VERY MINOR ethics concerns only"]

**Final Justification:**

The authors' rebuttal addressed some of my essential concerns. I believe that the presentation and clarity will be improved in the final version.

**Limitations:**

Yes.

**Quality:**

2

**Strengths And Weaknesses:**

Strengths
1. The paper is the first attempt to jointly solve defocus deblurring and dynamic scene deblurring in the monocular video-based dynamic Gaussian splatting construction.
2. The effectiveness of the proposal is backed by a sufficient number of experiments, increasing the credibility of the work.
3. The reported scores in the paper show outstanding performance in both types of blurs as addressed.

Weaknesses
1. The technical contributions are unclear in the manuscript. In Section 3, for example, the claimed contribution of dynamic Gaussian densification is mixed with other off-the-shelf tools like SAM, Depth-Anything, and TAPIR. It is difficult to discern which parts are the authors' original contributions.
2. Certain implementation details are omitted in the main manuscript. Hyperparameters like N_d, N_u remain as abstract symbols, making it more challenging to capture the details of how this method works in reality.
3. Ablation studies in Section 4.2 and the claimed contributions in Section 1 do not align. For example, it is unclear why BP-Net is ablated by only its shortcut connection if the usage of the network itself is claimed as a novelty.
4. Citations and references to previous methods and datasets are overly abstract. For example, Table 2 presents a crucial previous method simply as a number [62], requiring readers to click the link and search for the method it references. Another example is that all the datasets are presented in a “defocus blur dataset [30]” or “motion blur dataset [55]”.

Weaknesses are mainly about the presentation and lack of clarity in the claimed contributions, which I believe to be significant for a technical report. For suggestions and minor weaknesses, please refer to the Questions section.

---

> ### Author Rebuttal · Authors · 2025-07-31
>
> **_(Q1) Unclear clarification of technical contributions._**
>
> Our technical contributions are twofold: (i) we present the first unified framework for dynamic Gaussian Splatting from both defocused and motion-blurred monocular videos, (ii) we develop an effective blur prediction network and introduce a dynamic Gaussian densification as well as an unseen view combined scene optimization scheme. Note, we do not consider the off-the-shelf tools like SAM, Depth-Anything, and TAPIR as our contributions. We will clarify this in the paper to avoid possible misunderstandings.
>
> **_(Q2) Implementation details are omitted in the main manuscript._**
>
> Thank you for pointing this out. As described in the Paper Checklist, we provide the full implementation details in the supplementary material. We will move the implementation details to the paper to facilitate reading and understanding.
>
> **_(Q3) Ablation study of the BP-Net._**
>
> Thank you for the constructive comment. In the table below, we conduct ablation study on the BP-Net, where we compare our method with two alternatives that replace BP-Net with the motion blur modeling approach introduced in Deblur4DGS and the defocus blur modeling method introduced in D2RF, respectively. As can be seen, our method with the proposed BP-Net produces better results than the other two alternatives, demonstrating the effectiveness of the BP-Net. We will include the ablation study in the paper.
>
> |Method||Defocus blur (D2RF [1])||||Motion blur (DyBluRF [2])||
> |-----|:--------:|:-----:|:-----:|:----:|:-----:|:----:|:----:|
> ||PSNR↑|SSIM↑|LPIPS↓||PSNR↑|SSIM↑|LPIPS↓|
> |Ours w/ motion blur in Deblur4DGS|28.63|0.829|0.094||26.74|0.859|0.060|
> |Ours w/ defocus blur in D2RF|28.96|0.832|0.094||26.30|0.825|0.109|
> |Ours with BP-Net|**29.39**|**0.859**|**0.078**||**27.01**|**0.876**|**0.056**|
>
> **_(Q4) Citations and references are overly abstract._**
>
> We will improve the citations and references according to the comment.
>
> **_(Q5) Comparing generated and estimated kernels to examine the effectiveness of BP-Net in predicting different types of blur._**
>
> Thanks for the comment. In the table below, we conduct comparison between generated and estimated kernels. To this end, we construct two datasets with ground truth blur kernels, referred to as D2RF-v2 and DyBluRF-v2, by randomly sampling two global Gaussian and linear distribution blur kernels of size 9 x 9 and then respectively applying them to the ground truth sharp images in D2RF [1] and DyBluRF [2] to obtain the corresponding blurry images. With the two datasets, we quantitatively compare our estimated blur kernels and the ground truth blur kernels using PSNR and KL divergence as metrics. As shown below, our estimated blur kernels are highly similar to the ground truth blur kernels in numerical metrics, manifesting the effectiveness of the BP-Net in predicting different types of blur. We will add the table and also visual comparison of blur kernels to the supplementary material.
>
> |       |||  D2RF-v2  || | DyBluRF-v2 |
> | ---- | :-----: | :----: |:-----: |:----: |:-----: |:----: |
> | PSNR↑    ||| 32.516|| | 29.941 |
> | KL Div.↓ ||| 0.214 | ||0.247  |
>
> **_(Q6) What if the two types of blurs occur jointly? Can the proposed method also handle this type of degradation?_**
>
> Thank you for the insightful comment. In the table below, we evaluate our method on two datasets (DyBluRF-v3 and D2RF-v3) with both motion and defocus blur occurring simultaneously. Note, we obtain the DyBluRF-v3 dataset by applying depth-of-field (DoF) rendering technique in Bokehme [3] to the original DyBluRF dataset [2] to simulate defocus blur, and create the D2RF-v3 dataset with motion blur by processing the original D2RF dataset [1] using the motion blur generation method in Davanet [4]. As shown, our method clearly outperforms all the compared methods on the two datasets, verifying the advantage of our method in handling cases with motion and defocus blur occurring jointly.
>
> |Method||D2RF-v3||||DyBluRF-v3||
> |-------|:-----:|:-----:|:-----:|:-----:|:------:|:-----:|:-----:|
> ||PSNR↑|SSIM↑|LPIPS↓||PSNR↑|SSIM↑|LPIPS↓|
> |D3DGS|23.66|0.739|0.257||21.75|0.655|0.289|
> |SoM|29.04|0.820|0.094||27.28|0.791|0.103|
> |D2RF|27.82|0.795|0.132||25.89|0.722|0.133|
> |DyBluRF|27.30|0.771|0.150||26.54|0.753|0.112|
> |De4DGS|29.74|0.856|0.078||27.97|0.824|0.087|
> |Ours|**30.26**|**0.885**|**0.062**||**28.55**|**0.859**|**0.064**|
>
> **_References_**
>
> [1] Luo, X., Sun, H., Peng, J., & Cao, Z. Dynamic neural radiance field from defocused monocular video. ECCV 2024.
>
> [2] Sun, H., Li, X., Shen, L., Ye, X., Xian, K., & Cao, Z. Dyblurf: Dynamic neural radiance fields from blurry monocular video. CVPR 2024.
>
> [3] Peng, J., Cao, Z., Luo, X., Lu, H., Xian, K., & Zhang, J. Bokehme: When neural rendering meets classical rendering. CVPR 2022.
>
> [4] Zhou, S., Zhang, J., Zuo, W., Xie, H., Pan, J., & Ren, J. S. Davanet: Stereo deblurring with view aggregation. CVPR 2019.

---

> > ### Author Response · Authors · 2025-08-03
> >
> > Dear Reviewer ocL4,
> >
> > Thank you again for your time and valuable feedback on our paper. We've carefully addressed all your concerns in our rebuttal and hope these clarifications sufficiently address them. If you have any follow-up questions or require additional details, we are very happy to discuss further.
> >
> > Best regards,
> >
> > The authors

---

### Official Review · Reviewer_XJhL · 2025-07-03

**Clarity:** 3
**Significance:** 3
**Originality:** 2
**Rating:** 5
**Confidence:** 4

**Summary:**

This paper introduces a framework for dynamic Gaussian Splatting capable of synthesizing high-quality, sharp novel views from monocular videos affected by either defocus or motion blur. The approach builds on blur kernel-based convolution to handle both blur types within a unified process. Key contributions include a blur prediction network (BP-Net) that predicts per-pixel blur kernels and blur intensity, a blur-aware sparsity constraint to improve kernel reliability, and a dynamic Gaussian densification strategy to address incomplete scene regions. Additionally, the method renders and constrains adjacent unseen views to prevent geometry from overfitting. Extensive experiments on benchmark datasets demonstrate superior quantitative and qualitative performance compared to recent baselines for both defocus and motion-blurred videos.

**Questions:**

Please refer to the weaknesses part. I will adjust my ratings according to the authors' response.

**Ethical Concerns:**

["NO or VERY MINOR ethics concerns only"]

**Final Justification:**

The authors' thorough rebuttal addressed most of my concerns. In the rebuttal, the authors demonstrated the robustness and performance, validated the unified model, and clarified the limitations and computational complexity.

**Limitations:**

yes.

**Quality:**

3

**Strengths And Weaknesses:**

## Strengths
- This work is the first 4DGS reconstruction method that combines motion and defocus blur handling into a single end-to-end pipeline
- The architecture and framework are clearly described
- The evaluation is comprehensive, covering both defocus and motion blur, across many SOTA methods

## Weaknesses
- Minimal Improvement: The numerical increase in the quantitative evaluation seems small, and motion blur still exists in most of the provided videos
- External Reliance on Preprocessing and Segmentation: The initialization for both static and dynamic Gaussians depends heavily on external methods for depth estimation, 2D point tracking, and SAM. There is no analysis provided on how errors from these preprocessing steps propagate into the final results, nor is the robustness to their failure modes systematically evaluated.
- Effectiveness of the Unified Blur Model: While the unification of defocus and motion blur via a kernel-based convolution is mathematically plausible, there is relatively little discussion of cases where these two blur types may not be accurately captured by the same convolutional abstraction—especially in the presence of spatially and temporally varying blurs with complex overlap. To this end, the unified model seems to be a naive approach - will it be more effective and easier to train if the motion and defocus blur are estimated separately (e.g. deblurring 3d gaussian splatting, ECCV 2024)? I would suggest the authors can first evaluate on some static blurry datasets.
- Limited Analysis of Computational Costs: Table 1 reports parameters, training time, and FPS for a single configuration, but there is no detailed analysis of training/inference scalability, memory demands, or sensitivity to scene complexity/sequence length. Furthermore, there is no breakdown of where computational bottlenecks occur (e.g., BP-Net, rasterization, kernel prediction, Gaussian densification).
- Lacks Comparison with MoBGS: Motion Deblurring Dynamic 3D Gaussian Splatting for Blurry Monocular Video

---

> ### Author Rebuttal · Authors · 2025-07-31
>
> **_(Q1)  The numerical increase in the quantitative evaluation seems small._**
>
> As shown in Table 1 in the paper, our numerical performance improvement over current SOTA method (Deblur4DGS) is evident on the PSNR$\uparrow$/SSIM$\uparrow$/LPIPS$\downarrow$ metrics (defocus blur: 29.39/0.859/0.078 vs. 28.49/0.791/0.154, motion blur: 27.01/0.876/0.056 vs. 26.62/0.871/0.059), especially considering that our method has significantly fewer parameters than Deblur4DGS (192.2M vs. 754.6M). Moreover, we would like to point out that, our work makes the first attempt towards developing a unified framework for dynamic Gaussian Splatting from both defocused and motion-blurred monocular videos, which we believe is more important than the performance improvement.
>
> **_(Q2) Motion blur still exists in the provided videos._**
>
> Despite the advantage over the compared methods, some of our provided videos do suffer from weak motion blur, because it remains a challenge for our method as well as other SOTA methods to achieve perfect dynamic object reconstruction for complex cases, which is also illustrated in our limitation analysis in Figure 8 in the paper.
>
> **_(Q3) More analysis on how errors from external preprocessing steps (depth estimation, 2D point tracking, and SAM) affect the results._**
>
> Thank you for the suggestion. In the table below, we quantitatively evaluate how errors from external preprocessing steps affect the robustness of our method. To simulate errors from depth estimation, we randomly scale and shift the estimated depth maps within the range of [0.8, 1.2] and [-20, 20], respectively. To simulate errors from 2D point tracking and SAM, we randomly shift the 2D tracking points within the range of [-30, 30], and randomly add or delete five 25 X 25 mask regions towards the mask predicted by SAM. As shown, our results produced with the artificially perturbed depth, tracking points, and motion mask are comparable to those produced with the originally estimated depth, tracking points, and motion mask, indicating that our method has some tolerance to errors from external preprocessing steps.
> |Method||Defocus blur (D2RF [1])||||Motion blur (DyBluRF [2])||
> |-----|:-----:|:-----:|:----:|:-----:|:----:|:----:|:----:|
> ||PSNR↑|SSIM↑|LPIPS↓||PSNR↑|SSIM↑|LPIPS↓|
> |Ours w/ depth perturbation|29.19|0.844|0.088||26.79|0.862|0.074|
> |Ours w/ tracking perturbation|29.21|0.854|0.084||26.86|0.874|0.060|
> |Ours w/ mask perturbation|29.25|0.854|0.085||26.74|0.865|0.067|
> |Ours|**29.39**|**0.859**|**0.078**||**27.01**|**0.876**|**0.056**|
>
> **_(Q4) More discussion on cases where defocus and motion blur may not be accurately captured by the same convolutional abstraction._**
>
> For complex cases that involve fast-moving objects in defocused background, the convolutional abstraction may struggle to accurately model defocus and motion blur. We will add some examples of such cases to the supplementary material.
>
> **_(Q5) Will it be more effective and easier to train if the motion and defocus blur are estimated separately (e.g. deblurring 3d gaussian splatting, ECCV 2024)?_**
>
> We have followed your suggestion to train our method by separately estimating motion and defocus blur as done in the mentioned ECCV 2024 paper [3]. As shown in the table below, adopting the separate modeling strategy results in clear performance drop on all the three metrics, manifesting the effectiveness of our unified modeling of motion and defocus blur. We will include the table in the paper.
>
> |Method||Defocus blur (D2RF [1])||||||Motion blur (DyBluRF [2])||
> |------|:-----:|:-------:|:---:|:-----:|:---------:|:------:|:------:|:------:|:------:|
> ||PSNR↑|SSIM↑|LPIPS↓||||PSNR↑|SSIM↑|LPIPS↓|
> |Ours w/ separate modeling|28.31|0.812|0.098||||26.05|0.823|0.118|
> |Ours|**29.39**|**0.859**|**0.078**||||**27.01**|**0.876**|**0.056**|
>
> **_(Q6) More evaluation on some static blurry datasets._**
>
> Thank you for the suggestion. In the table below, we compare our method with a recent method (Deblurring 3D Gaussian Splatting [3], ECCV 2024) tailored for static blurry images on the Deblur-NeRF dataset [4]. As shown, our method consistently outperforms the compared method (referred to as Deblurring-GS) on both motion and defocus blur, validating the effectiveness of our method in dealing with static blurry images. We will add the table to the paper.
>
> |Method||Defocus blur (Deblur-NeRF [4])||||Motion blur (Deblur-NeRF [4])||
> |---|---|:---:|:---:|:---:|:---:|:---:|:---:|
> ||PSNR↑|SSIM↑|LPIPS↓||PSNR↑|SSIM↑|LPIPS↓|
> |Deblurring-GS|23.71|0.747|0.110||26.61|0.822|0.108|
> |Ours|**24.22**|**0.768**|**0.095**||**27.14**|**0.835**|**0.096**|
>
> **_(Q7) More analysis on computational cost and memory demands._**
>
> Our model requires an average of 6GB GPU memory for both training and inference. During training, the BP-Net dominates the computational cost with a ratio of 40%, while the dynamic Gaussian deformation and rasterization respectively occupy 25% and 15% of the computational cost, and the rest of 10% is for loss computation. During inference, the computational cost only comes from dynamic Gaussian deformation and rasterization, with the former at 70% and the latter at 30%.
>
> **_(Q8) Comparison with MoBGS._**
>
> Thank you for the comment. Since the implementation of MoBGS [5] is not yet publicly available, we compare our method with MoBGS by training on the same motion blur dataset as MoBGS in terms of PSNR$\uparrow$/SSIM$\uparrow$/LPIPS$\downarrow$ on the full region. The comparative results between our method and MoBGS are 28.61/0.942/0.059 vs. 28.70/0.945/0.051 (the results of MoBGS are from their paper), showing that our method can produce comparable results to MoBGS on motion blur, even though MoBGS is specially designed for motion blur while our method can effectively handle both motion and defocus blur.
>
>
> **_References_**
>
> [1] Luo, X., Sun, H., Peng, J., & Cao, Z. Dynamic neural radiance field from defocused monocular video. ECCV 2024.
>
> [2] Sun, H., Li, X., Shen, L., Ye, X., Xian, K., & Cao, Z. Dyblurf: Dynamic neural radiance fields from blurry monocular video. CVPR 2024.
>
> [3] Lee, B., Lee, H., Sun, X., Ali, U., & Park, E. Deblurring 3d gaussian splatting. ECCV 2024.
>
> [4] Ma, L., Li, X., Liao, J., Zhang, Q., Wang, X., Wang, J., & Sander, P.V. Deblur-nerf: Neural radiance fields from blurry images. CVPR 2022.
>
> [5] Bui, M. Q. V., Park, J., Bello, J. L. G., Moon, J., Oh, J., & Kim, M. MoBGS: Motion Deblurring Dynamic 3D Gaussian Splatting for Blurry Monocular Video. arXiv 2025.

---

> > ### Author Response · Authors · 2025-08-03
> >
> > Dear Reviewer XJhL,
> >
> > Thank you again for your time and valuable feedback on our paper. We've carefully addressed all your concerns in our rebuttal and hope these clarifications sufficiently address them. If you have any follow-up questions or require additional details, we are very happy to discuss further.
> >
> > Best regards,
> >
> > The authors

---

> > > ### Comment · Reviewer_XJhL · 2025-08-04
> > >
> > > Dear Authors,
> > >
> > > Thank you for your thorough rebuttal and clarifications. Most of my concerns have been addressed. I value your detailed responses and have adjusted my evaluation accordingly.
> > >
> > > Best regards,
> > > Reviewer XJhL

---

### Decision · Program_Chairs · 2025-09-17

**Decision:**

Accept (poster)

**Comment:**

The paper proposes a deblurring method based on Gaussian splatting that simultaneously addresses both defocus and motion blur. Although the paper initially received three negative ratings, the rebuttal effectively clarified key details and convinced the reviewers with additional qualitative results. All reviewers acknowledge the novelty on the joint formulation framework, comprehensive evaluations, and improved clarity. The AC concurs with the reviewers and recommends clear acceptance. The authors are encouraged to improve the paper presentation according to the suggestions from reviewers ocL4, qGyt, and Uuup when preparing the camera ready version.